# An Unusual Presentation of Crohn’s Disease Diagnosed Following Accidental Ingestion of Fruit Pits: Report of Two Cases and Review of the Literature

**DOI:** 10.3390/life11121415

**Published:** 2021-12-16

**Authors:** Emanuele Sinagra, Dario Raimondo, Salvatore Marco Iacopinelli, Francesca Rossi, Giuseppe Conoscenti, Maria Angela Di Maggio, Sergio Testai, Rita Alloro, Marta Marasà, Alberto Calandra, Claudia Costanza, Serena Cristofalo, Socrate Pallio, Marcello Maida, Ilaria Tarantino, Goffredo Arena

**Affiliations:** 1Department of Endoscopy, Fondazione Istituto G. Giglio, Contrada Pietra Pollastra Pisciotto, 90015 Cefalù, Italy; dario.raimondo@hsrgiglio.it (D.R.); fraross76@hotmail.com (F.R.); dottgconoscenti@gmail.com (G.C.); ritalloro@hotmail.com (R.A.); 2Department of Surgery, Fondazione Istituto G. Giglio, Contrada Pietra Pollastra Pisciotto, 90015 Cefalù, Italy; marksurg@libero.it (S.M.I.); mariaangela.dimaggio@hsrgiglio.it (M.A.D.M.); goffredoarena@gmail.com (G.A.); 3Department of Radiology, Fondazione Istituto G. Giglio, Contrada Pietra Pollastra Pisciotto, 90015 Cefalù, Italy; sergio.testai@virgilio.it (S.T.); martamarasa@libero.it (M.M.); alberto.calandra@hsrgiglio.it (A.C.); claudia.costanza@hsrgiglio.it (C.C.); serena.cristofalo@hsrgiglio.it (S.C.); 4Endoscopy Service, Department of Clinical and Experimental Medicine, University of Messina, AOUP Policlinico G. Martino, 98125 Messina, Italy; socratep@tin.it; 5Gastroenterology and Endoscopy Unit, S. Elia-Raimondi Hospital, 93100 Caltanissetta, Italy; marcello.maida@hotmail.it; 6Endoscopy Service, Department of Diagnostic and Therapeutic Services, IRCCS-ISMETT, 90127 Palermo, Italy; itarantino74@gmail.com; 7Department of Surgery, McGill University, Montreal, QC H3G 1A4, Canada; 8Istituto Oncologico del Mediterraneo, 95029 Viagrande, Italy

**Keywords:** Crohn’s disease, intestinal strictures, fruit pit, bowel obstruction, bezoars

## Abstract

The clinical course of Crohn’s disease (CD) is often complicated by intestinal strictures, which can be fibrotic, inflammatory, or mixed, therefore leading to stenosis and eventually symptomatic obstruction. We report two cases of subclinical CD diagnosed after fruit pit ingestion, causing bowel obstruction; additionally, we conducted a narrative review of the scientific literature on cases of intestinal obstruction secondary to impacted bezoars due to fruit pits. Symptoms of gastrointestinal bezoars in CD patients are not diagnostic; and the diagnosis should be based on a combined assessment of history, clinical presentation, imaging examination and endoscopy findings. This report corroborates the concept that CD patients are at a greater risk of bowel obstruction with bezoars generally and shows that accidental ingestion of fruit pits may lead to an unusual presentation of the disease. Therapeutic options in this group of patients differ from the usual approaches implemented in other patients with strictures secondary to CD.

## 1. Introduction

The clinical course of Crohn’s disease (CD) is often complicated by intestinal strictures, which can be fibrotic, inflammatory, or mixed, therefore leading to stenosis and ultimately to symptomatic obstruction [1].

The presence of strictures in CD ranges from 12 to 54%, being most frequent in patients who have longstanding disease, and the terminal ileum is the most affected location [2,3,4,5]. The stricturing phenotype is characterized by the inflammatory and/or fibrotic luminal reduction of one or more segments of the gastrointestinal tract (GIT). Diagnosis can be made with clinical and endoscopic examinations, as well as with radiological and surgical findings [2,5].

A bezoar is defined as an indigestible accumulation that is trapped in the gastrointestinal tract. Such indigestible mass can be formed by a variety of materials that can be ingested intentionally or accidentally. Representative substances forming bezoars include plant materials such as fibers, skin or seeds of vegetables and fruits (i.e., phytobezoars), ingested hair (i.e., trichobezoars), medications (i.e., pharmacobezoars), and milk protein in milk-fed infants (i.e., lactobezoars) [6,7]. Bezoars can be formed and can be found in any part of the gastrointestinal tract, with the stomach being the most common location. Once the diagnosis of bezoar is made, the bezoar should be dissolved or removed because it can cause gastric outlet obstruction, ileus, ulcerations due to pressure necrosis, and subsequent gastrointestinal bleeding [6].

Small bowel obstruction can occur due to extramural, mural, or intraluminal causes, and adhesions account for more than 60%, followed by CD and malignancy [8]. Foreign bodies in the gastrointestinal tract can get obstructed depending on their shape and consistency. Frequent sites of obstruction include the pylorus, ‘C’ loop of the duodenum, and the ileocaecal junction [8,9]. Usually, a foreign body that reaches the stomach has an 80–90% chance of passage [9]. Moreover, almost all pass through once it reaches the small bowel negotiating the pylorus and duodenal curve [9]. It rarely gets impacted at the ileocaecal valve. Time taken for natural passage is about 4–6 days, butrarely up to 4 weeks [9]. The likelihood of an ingested foreign body to pass through the anus depends on the size and shape [9]. Chance of passage is more when the width of the foreign body is less than 2.5 cm which passes through the pylorus, and when the length of the foreign body is less than 6 cm, which helps to negotiate the acute angles of the duodenal curve [9].

We report two cases of subclinical CD diagnosed after fruit pit ingestion causing bowel obstruction, and we conducted a narrative review of the scientific literature of the previous published cases of intestinal obstruction due to impacted bezoars.

### 1.1. Case 1

A 42-year-old man presented with a four-day history of colicky abdominal pain and vomiting. His past medical and surgical history were unremarkable. He denied taking any medication or any change in bowel habit.

On clinical examination, the abdomen was tender all over the quadrants, without any peritoneal signs elicited by manual palpation. Laboratory tests showed an increased white blood cell count of 12,000/mm^3^ and a C-reactive protein of 14.2 mg/dL. Computed tomography scan showed thickening of the terminal ileum with a 14 mm intraluminal radiopaque mass causing small bowel obstruction (Figure 1). Decision was taken to perform laparoscopic exploration of the abdomen for diagnostic and therapeutic purposes. During the laparoscopy, the thickened segment of small bowel was opened and a foreign body resembling a fruit pit was found within the inflamed lumen. The foreign body was extracted and closure of the small bowel was performed. Upon questioning the patient, after surgery, about the possibility of foreign body ingestion, it was discovered that the patient had accidentally swallowed a medlar pit a couple of days earlier. A month later a colonoscopy was performed, which confirmed a stenosis of the ileocecal valve suspicious for inflammatory bowel disease (Figure 2). Pathology examination of the biopsy specimens confirmed the diagnosis of CD involving the terminal ileum and the ileocecal valve. Due to the nature of the stenosis, which was fibrotic and stenotic, an ileocecal resection was planned and performed successively without any complications.

### 1.2. Case 2

A 58-year-old woman presented to emergency room with a three-day history of colicky abdominal pain and vomiting. Her medical history was consistent with hypertension and osteoporosis. She denied any change in bowel habit.

On clinical examination, the abdomen was tender all over the quadrants with absence of peritoneal signs. Laboratory tests showed an increased white blood cell count of 14,000/mm^3^ and a C-reactive protein of 26.2 mg/dl. Computed tomography scan showed small bowel thickening with a 12 mm intraluminal radiopaque mass (Figure 3). Colonoscopy showed an inflammatory (ulcerated) but serrated stenosis of the terminal ileum with patency of ileocecal valve, suspicious for ileal CD (Figure 4). Magnetic resonance imaging showed thickening of the distal ileum, 7 cm in length, without the presence of the fruit pit, that had been physiologically expelled from the colon (Figure 5). Due to the nature of the stenosis, which was inflammatory, a “wait and see approach” was chosen. After a brief course of systemic prednisone, which failed to obtain any clinical benefit, ileocecal resectionwas performed with resolution of all symptoms.

## 2. Discussion

Bowel strictures causing obstruction can be secondary to stricturing CD, postoperative adhesions [2,4,5], inflammatory parietal thickening [2,5], enterolytes [2,5,10,11,12,13,14], bezoars [2,5,15,16], seeds [2,5,17,18,19,20,21,22,23], medications in the form of pills [2,3,4], endometriosis [2,4], internal or incarcerarted hernias [2,4], intussusception [2,4], pseudopolyps [2,24], tumors [2,25], and gallstone ileus [2,4].

There are few case reports of bowel obstruction in patients with CD caused by bezoars. This condition is associated with many factors such as inadequate chewing of the food, teeth problems, a high-fiber diet, previous abdominal surgery, nonsteroidal anti-inflammatory drugs (NSAID), exposure, radiation, decreased motility, mental or visual impairment, and hyposecretion of the GIT [26]. In Table 1 there is a summary of all the available evidence on stricturing Crohn’s disease (CD) and bowel obstruction secondary to the ingestion of a fruit pit.

In the cases reported up to now, there is a prevalence of male sex. In three cases, treatment was performed through an open surgical approach (only in one of them through laparoscopy), whereas in only one case the treatment was performed endoscopically. 

The tendency to form intestinal strictures in CD patients may further facilitate the development of GI bezoars. CD is characterized by periods of clinical remission that can be followed by episodes of recurrence [26,27]. Although disease location in CD tends to be stable, disease behavior or phenotype may evolve over time. Henriksen et al. [26,28] showed that among 200 Norwegian patients with CD, 28% had a stricturing phenotype at the time of diagnosis. The rate of strictures went up to 33% after five-year follow-up. Change in disease behavior in a subset of patients with CD in the Olmsted County cohort, diagnosed between 1970 and 2004, was also described [26,29] In this cohort of patients, only 18.6% of the patients had evidence of stricturing or perforating intestinal complications at diagnosis or within 90 days of diagnosis. However, the cumulative risk of developing either complication was 50.8% at 20 years after diagnosis [26].

Fruit pit ingestion is an unusual cause of intestinal obstruction, and it is very rare that such an event leads to the diagnosis of CD [17]. Management of foreign body ingestion has been conservative, entailing close observation, as most objects pass through the gastrointestinal tract without any complications [15]. Intestinal obstruction and perforation are rare but potential risks and patients with intrinsic bowel disease are at a greater risk of developing symptoms [17,30]. Such complications are indications for urgent surgery and prompt removal of the foreign body [17].

Clinical manifestations of GI bezoars may vary, depending on the location, size and number of bezoars, and the presence or absence of bowel strictures [26]. Patients with GI bezoars may remain asymptomatic or develop life-threatening complications such as impaction, obstruction, perforation, and hemorrhage [26,31,32,33,34]. Common symptoms usually include abdominal pain or distension, nausea, vomiting, early satiety and weight loss.

A combined assessment of history, clinical presentation, imaging examination, and endoscopy is critical for the diagnosis [26,35,36] A well-taken history and physical examination may facilitate the diagnosis [26]. Imaging examinations such as abdominal radiography with or without contrast, and CT or MRI scan, where bezoars may be seen as a mass or a filling defect, are usually helpful [26]. Metal objects or calcified bezoars can easily be detected with radiographic imaging [26].

In the past, bezoars or foreign bodies in the GI tract, especially in patients with IBD, typically required surgical intervention [26]. However, due to the advancement of endoscopic equipment and techniques, a large proportion of patients with GI bezoars can be managed nowadays with endoscopic approaches [26].

Factors affecting the efficacy of the endoscopic therapy for bezoars and foreign bodies have not been systemically evaluated. Wu and Shen proposed that nature and location of bezoars or foreign bodies, bowel anatomy, the presence or absence of strictures, availability of endoscopic tools, and endoscopist’s expertise, play important roles [26]. From their experiences in the management of bezoars in ileal pouches and in strictured bowel, we found that lithobezoars, especially those of alarger size, are more likely to fail the endoscopic mechanical lithotripsy [26].

Simultaneous endoscopic therapy for the treatment of strictures in patients with IBDmay be a prerequisite for the subsequent retrieval of bezoars and foreign body. The most frequently employed technique is endoscopic balloon dilation (EBD) [26,37].

One of main disadvantages of EBD is the occurrence of blind tears, and lack of full control of the location, degree, and depth of dilation performed by the endoscopist [26,38].

Surgical removal should be reserved as the last therapeutic option for GI bezoars [26]. It is usually indicated for patients who fail conservative therapy or have large bezoars that hinder their endoscopic removal [26]. The type of surgical operation may be determined perioperatively or intraoperatively [15,26]. A closed technique, which requires manual cracking of bezoars and pushing through the site of stricture, can be attempted [24]. Enterotomy, albeit invasive, is more effective. The enterotomy should be performed at a site distant from the site of the impaction to prevent intestinal leaks due to an acutely inflamed mucosa and subsequent impaired healing. The entire GI tract should be explored with caution, since there is 4% chance of multiple bezoars [15,26]. Therefore, a diligent search for additional strictures with stricturoplasty during the same operation is advised. If stricturoplasty is not feasible, bowel-sparing resection can be performed [24]. Laparoscopy has been proven to be safe in experienced hands for patients with CD. The laparoscopic approach is associated with a shorter hospital stay, faster recovery, and better cosmesis than the open approach [26].

Wu and Chen recently proposed a management algorithm for bezoars and foreign bodies in inflammatory bowel disease (IBD) [26]. According to these authors, in case of asymptomatic or partial small bowel obstruction, with inflammatory stricture, they suggest a “wait and see” approach and medical treatment with steroids; in case of failure of the conservative management, an endoscopic or surgical management is strongly advised [26]. For our second patient, due to the inflammatory nature of the stenosis, a “wait and see approach” with systemic prednisone was chosen. Owing to the lack of any clinical benefit, an ileocaecal resectionwas planned with complete resolution of the symptoms.

In the case of partial small bowel obstruction secondary to a fibrotic stricture, an endoscopic retrieval of the foreign body with an associated endoscopic treatment of the stricture may be considered as the first option; in the case of failure, surgical management would be the alternative [26]. On the other hand, in cases of complete small bowel obstruction, haemorrhage, or perforation, surgery must be considered the treatment of choice. That was the strategy used in our first patient, where, due to the nature of the stenosis that was fibrotic and stenotic, a laparoscopic ileocecal resection was planned and performed successively without any complications.

In conclusion, since symptoms of GI bezoars in patients with IBD are not diagnostic, the diagnosis of such disease should be based on a combined assessment of history, clinical presentation, imaging examination, and endoscopy findings [26,39,40,41].

This report corroborates the concept that CD patients are at a greater risk of bowel obstruction with bezoars generally and shows that accidental ingestion of fruit pits may lead to an unusual presentation of the disease. Therapeutic options in this group of patients differ from the usual approaches implemented in other patients with strictures secondary to CD. 

## Figures and Tables

**Figure 1 life-11-01415-f001:**
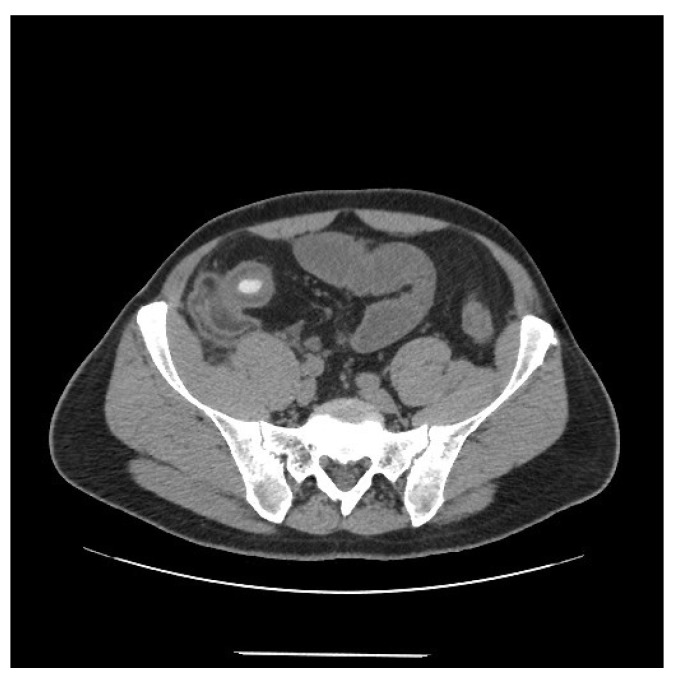
Computed tomography scan showing thickening of the terminal ileum with a 14 mm intraluminal radiopaque mass.

**Figure 2 life-11-01415-f002:**
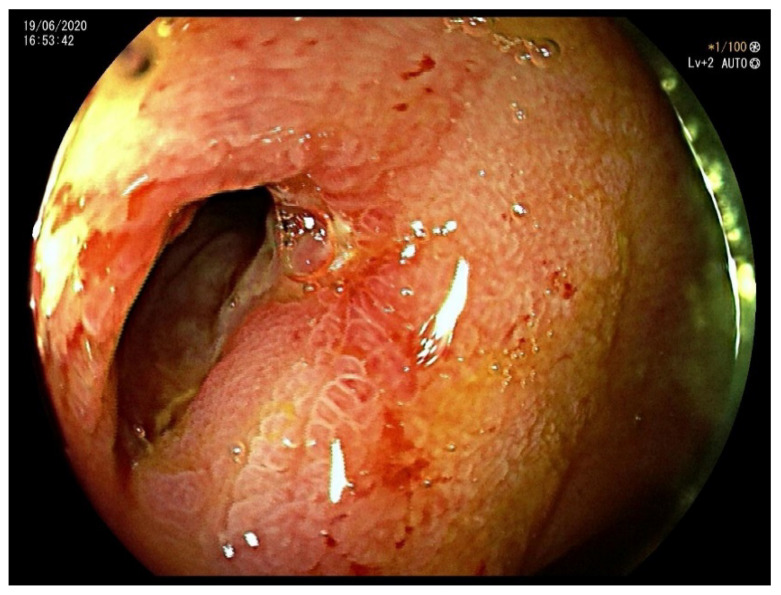
Colonoscopy showing a stenosis of the ileocecal valve suspicious for inflammatory bowel disease.

**Figure 3 life-11-01415-f003:**
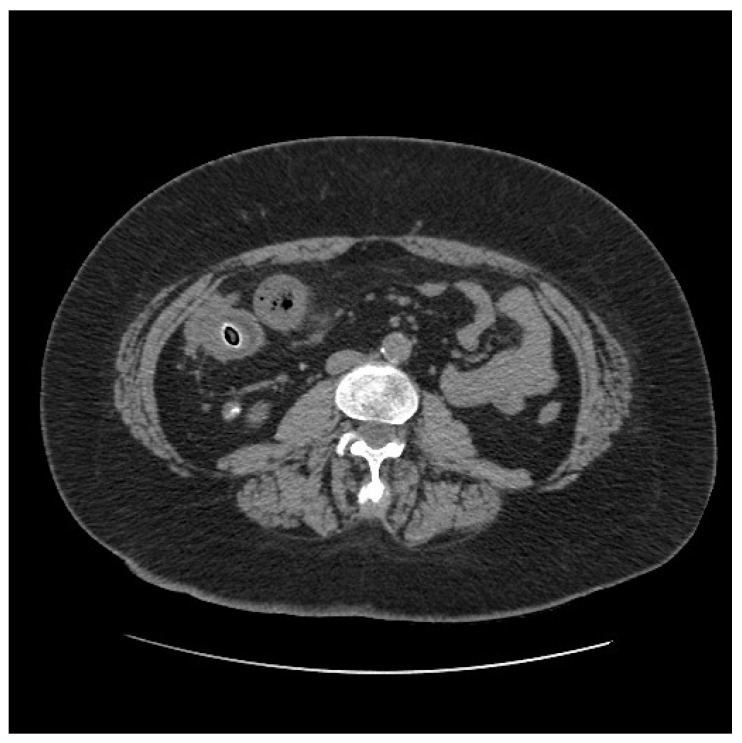
Computed tomography scan showing small bowel thickening with a 12 mm intraluminal radiopaque mass.

**Figure 4 life-11-01415-f004:**
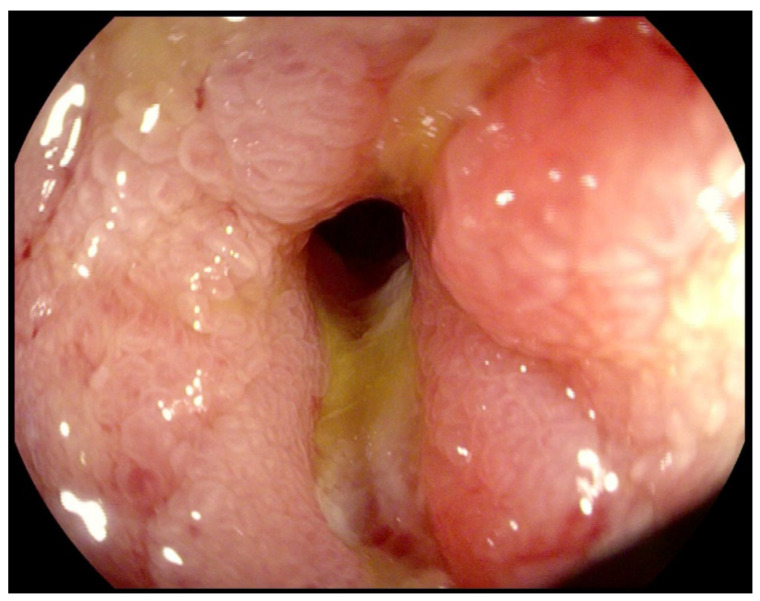
Colonoscopy showing an inflammatory (ulcerated) but serrated stenosis of the terminal ileum with patency of the ileocecal valve.

**Figure 5 life-11-01415-f005:**
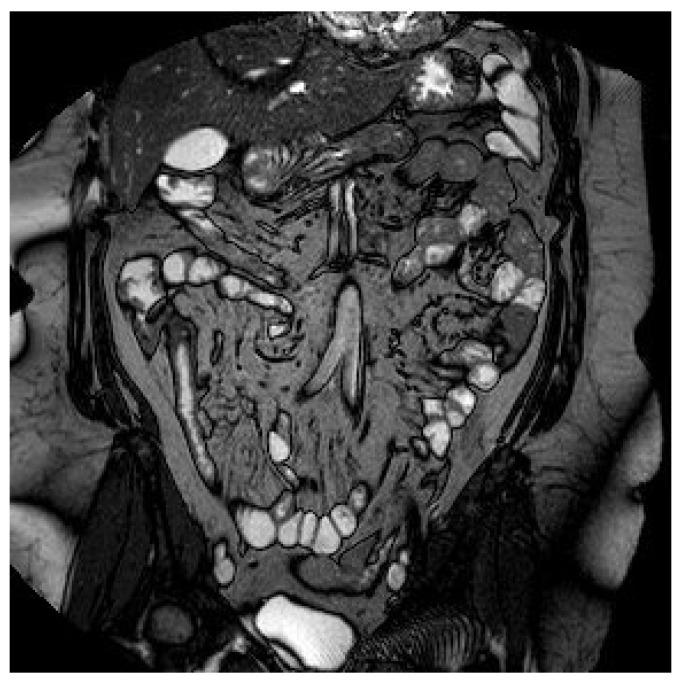
Magnetic resonance imaging showing a thickening of the distal ileum, extended for 7 cm, with contrast enhancement but without the presence of the fruit pit.

**Table 1 life-11-01415-t001:** Literature data about fruit pit ingestion causing small bowel obstruction in Crohn’s disease.

Author [ref.]	Year of Publication	Patient’s Age	Patient’s Sex	Type of Seeds	Management
Kaufman, D [15]	2001	62	Male	Plum pit	Surgery (Laparotomy)
Shedda, S [16]	2006	45	Male	Olive pit	Surgery (Laparotomy)
Slim, R [17]	2006	55	Female	Fruit pit	Surgery (Laparoscopy)
Garau, MV [19]	2009	48	Male	Medlar pit	Endoscopy (Balloon dilation and retrieval)

## Data Availability

Not applicable.

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
