# Peer review of "An Unusual Presentation of Crohn’s Disease Diagnosed Following Accidental Ingestion of Fruit Pits: Report of Two Cases and Review of the Literature"

_life, 2021, doi:10.3390/life11121415_

Round 1
Reviewer 1 Report
- Define the abbreviations the first time they appear (for example, “ER”)
- Figure 1 and Figure 3 seem identical
- Why patient 1 underwent surgery before colonoscopy and patient 2 perform immediate colonoscopy? How did you prepare a patient with occlusion?
- What is a “ileotyphlectomy”?
- Since patient two was treated with steroids, did you perform temporary ostomy?
- Use oxford comma in the whole text
- “The most valuable diagnostic and therapeutic procedure for bezoars detection is gastrointestinal endoscopy, which can provide both direct visualization of the bezoar and therapeutic intervention”
Is this true for both gastric and ileal bezoars?
- “IBD patients”
Write “patients with IBD”
- “Retrograde endoscopic removal of GI bezoars or foreign bodies in IBD patients usually involves fragmenting soft bezoars with water jet, forceps, or snares”
How do you reach a bezoar in the ileum of a patients with an inflamed, more than 5 cm, stenosis of the terminal ileum?
- “Wu and Shen developed needle-knife stricturotomy (NKSt) as a novel endoscopic technique for the treatment of strictures in the patients with IBD”
This is neither real life, nor present in guidelines
- “Surgical removal should be reserved as the last therapeutic option for GI bezoars”
If this is true, why your two cases both underwent surgery in a few weeks?
- “According to these authors, in case of asymptomatic or partial small bowel obstruction, with inflammatory stricture, they suggest a “wait and see” approach and medical treatment with steroids and biologic agents”
Using biological therapy in patients with stenosing CD is absolutely contraindicated
Author Response
Dear Editors, dear Reviewers,
We wish to express our appreciation to the Editors and Reviewers for their insightful comments, which have helped us to improve our manuscript significantly. According to the suggestions, we have thoroughly revised our manuscript and its final version is enclosed. Point-by-point responses to the comments are listed below.
Reviewers’ comments #1
“Define the abbreviations the first time they appear (for example, “ER”)”
Response: We thoroughly checked the manuscript and we made corrections as the reviewer suggested.
“Figure 1 and Figure 3 seem identical”
Response: We apologize for the mistake and we thank the reviewer for pointing that out. We uploaded the correct image in figure 3.
“Why patient 1 underwent surgery before colonoscopy and patient 2 perform immediate colonoscopy? How did you prepare a patient with occlusion?”
Response: Patient 1 underwent surgery because he presented with a complete bowel obstruction. In the discussion we stressed that “in cases of complete small bowel obstruction, haemorrhage or perforation, surgery must be considered the treatment of choice. That is why the strategy used in our first patient was surgical.
We added to the text: “Computed tomography scan showed thickening of the terminal ileum with a 14 mm intraluminal radiopaque mass causing small bowel obstruction (Figure 1).”
Patient 2, presented with abdominal pain without symptoms of bowel obstruction. The vomit didn’t have any enteric content but only gastric fluid. Therefore a “wait and see approach” with systemic prednisone was chosen as the available literature suggests in these cases.
The patients was prepped through a nasogastric tube and colonoscopy was performed to try and retrieve the fruit pit endoscopically.
“What is a “ileotyphlectomy”
Response: We apologize for the mistake. We meant ileocecal resection and therefore we corrected the mistake by substituting this term with “ileocecal resection” throughout the manuscript.
“Since patient two was treated with steroids, did you perform temporary ostomy?”
Response: We did not perform temporary ostomy, since we performed a steroid tapering approach with an appropriate steroid wash out (2-week) before surgery.
“The most valuable diagnostic and therapeutic procedure for bezoars detection is gastrointestinal endoscopy, which can provide both direct visualization of the bezoar and therapeutic intervention - Is this true for both gastric and ileal bezoars?”
Response: Since this sentence may be confusing, due to the fact that gastrointestinal endoscopy is more useful in gastric rather than ileal bezoars, we erased the sentence. Thank you for the useful comment.
“IBD patients” - Write “patients with IBD”
Response: we made changes throughout the manuscript according to the reviewer suggestion
“Retrograde endoscopic removal of GI bezoars or foreign bodies in IBD patients usually involves fragmenting soft bezoars with water jet, forceps, or snares”
“How do you reach a bezoar in the ileum of a patients with an inflamed, more than 5 cm, stenosis of the terminal ileum?”
Response : We agree with the rewiever that our statement is out of context and therefore we deleted it
“Wu and Shen developed needle-knife stricturotomy (NKSt) as a novel endoscopic technique for the treatment of strictures in the patients with IBD” - This is neither real life, nor present in guidelines”
Response: We mentioned this option as a novel treatment and it was meant to be taken as such. We agree with the reviewer that maybe this treatment is still too novel and needs further validation with clinical trials before being suggested as a possible option in these cases. For these reasons we erased this statement from the manuscript.
“Surgical removal should be reserved as the last therapeutic option for GI bezoars”
If this is true, why your two cases both underwent surgery in a few weeks?
Response: The first case underwent surgery because of a complete mechanical small bowel obstruction in a virgin abdomen. The second case went to surgery only after failure of the conservative approach. Had the medical treatment worked, we would have avoided the surgical option.
“According to these authors, in case of asymptomatic or partial small bowel obstruction, with inflammatory stricture, they suggest a “wait and see” approach and medical treatment with steroids and biologic agents” - Using biological therapy in patients with stenosing CD is absolutely contraindicated
Response: We agree with the reviewer and we apologize for the gross mistake.
Reviewer 2 Report
Emanuele Sinagra and colleagues present a quality and well-written case report manuscript describing an unusual presentation of Crohn’s disease diagnosed following accidental ingestion of fruit pits.
Authors report two cases of subclinical Crohn’s disease diagnosed after fruit pit ingestion causing bowel obstruction. They also provide an overview of the scientific literature of the previous published cases of intestinal obstruction due to impacted bezoars.
Authors suggest that symptoms of gastrointestinal tract bezoars in Crohn’s disease patients are not diagnostic and the diagnosis should be based on a combined assessment of patient’s history, clinical presentation, imaging examination and endoscopy findings.
Finally, authors conclude that this case report corroborates the concept that Crohn’s disease patients are at a greater risk of bowel obstruction with bezoars generally and show that accidental ingestion of fruit pits may lead to an unusual presentation of the disease.
Other comments:
1) Please check for typos throughout the manuscript.
2) Please provide full term for IBD for the first appearance of this word. Add “inflammatory bowel disease”.
3) Authors are kindly encouraged to cite the following article that describes certain aspects of autoimmune disorders (like Crohn’s disease). DOI: 10.1007/s12668-016-0233-x
Overall, the manuscript is highly valuable for the scientific community and should be accepted for publication.
Author Response
Response to reviewers’ comments
Dear Editors, dear Reviewers,
We wish to express our appreciation to the Editors and Reviewers for their insightful comments, which have helped us to improve our manuscript significantly. According to the suggestions, we have thoroughly revised our manuscript and its final version is enclosed. Point-by-point responses to the comments are listed below.
Reviewers’ comments #2
Emanuele Sinagra and colleagues present a quality and well-written case report manuscript describing an unusual presentation of Crohn’s disease diagnosed following accidental ingestion of fruit pits.
Authors report two cases of subclinical Crohn’s disease diagnosed after fruit pit ingestion causing bowel obstruction. They also provide an overview of the scientific literature of the previous published cases of intestinal obstruction due to impacted bezoars.
Authors suggest that symptoms of gastrointestinal tract bezoars in Crohn’s disease patients are not diagnostic and the diagnosis should be based on a combined assessment of patient’s history, clinical presentation, imaging examination and endoscopy findings.
Finally, authors conclude that this case report corroborates the concept that Crohn’s disease patients are at a greater risk of bowel obstruction with bezoars generally and show that accidental ingestion of fruit pits may lead to an unusual presentation of the disease.
Other comments:
- Please check for typos throughout the manuscript.
Response:
We would like to thank the reviewer for the kind words used to describe the manuscript. The manuscript, as requested, was revised by a native English speaker from MDPI and all the typos corrected.
2) Please provide full term for IBD for the first appearance of this word. Add “inflammatory bowel disease”.
Response: Thank you for the suggestion. We corrected this item as suggested.
3) Authors are kindly encouraged to cite the following article that describes certain aspects of autoimmune disorders (like Crohn’s disease). DOI: 10.1007/s12668-016-0233-x
Response: As suggested we added the aforementioned reference
Overall, the manuscript is highly valuable for the scientific community and should be accepted for publication.
Reviewer 3 Report
Sinagra et al. present two cases of presentation of Crohn’s disease diagnosed following accidental ingestion of fruit pits. I have the following remarks:
The introduction section may be expanded with addressing the problem in light of differential diagnosis.
Line 74. no need to define CD twice
Line 95 Hers
Figure 1 and Figure 3 appear the same, are the authors sure that they provided the correct figure.
Line 123. This kind of statistical analysis is redundant as this sample size is to small to make any generalizations, please omit it.
Finally, although quite interesting, I don’t see any significance of these two cases for the available literature, as the underlying mechanisms are well-established, and these conditions are too rare to be considered in everyday practice. In fact, neither the authors did not had it in mind until they saw CT results, and in patient with history of colicky abdominal pain and vomiting of unknown cause, alongside indices of inflammation, CT would probably be performed as a part of standard approach.
Author Response
Response to reviewers’ comments
Dear Editors, dear Reviewers,
We wish to express our appreciation to the Editors and Reviewers for their insightful comments, which have helped us to improve our manuscript significantly. According to the suggestions, we have thoroughly revised our manuscript and its final version is enclosed. Point-by-point responses to the comments are listed below.
Reviewers’ comments #3
Sinagra et al. present two cases of presentation of Crohn’s disease diagnosed following accidental ingestion of fruit pits. I have the following remarks:
“The introduction section may be expanded with addressing the problem in light of differential diagnosis.”
Response: We would like to thank the reviewer for the valuable suggestion. We expanded the introduction and now it can be read:
- "Small bowel obstruction can occur due to extramural, mural or intraluminal causes, and adhesions account for more than 60%, followed by CD and malignancy [8]. Foreign bodies in the gastrointestinal tract can get obstructed depending on their shape and consistency. Frequent sites of obstruction include the pylorus, ‘C’ loop of the duodenum and the ileocaecal junction [8,9]. Usually, foreign body that reaches the stomach has an 80–90% chance of passage [9]. And almost all pass through, once it reaches the small bowel negotiating the pylorus and duodenal curve [9]. It rarely gets impacted at the ileocaecal valve. Time taken for natural passage is about 4–6 days and rarely up to 4 weeks [9]. The likelihood of an ingested foreign body to pass out through the anus depends on the size and shape [9]. Chance of passage is more when the width of the foreign body is less than 2.5 cm which passes through the pylorus and when the length of the foreign body is less than 6 cm, which helps to negotiate the acute angles of the duodenal curve [9].”
Line 74. no need to define CD twice
Response: we corrected the sentence as suggested
Line 95 Hers
Response: we corrected the sentence as suggested
Figure 1 and Figure 3 appear the same, are the authors sure that they provided the correct figure.
Response: We apologize for the mistake and we thank the reviewer for pointing that out. We uploaded the correct image in figure 3.
Line 123. This kind of statistical analysis is redundant as this sample size is to small to make any generalizations, please omit it.
Response :
We erased the statistical analysis as suggested.
“Finally, although quite interesting, I don’t see any significance of these two cases for the available literature, as the underlying mechanisms are well-established, and these conditions are too rare to be considered in everyday practice. In fact, neither the authors did not had it in mind until they saw CT results, and in patient with history of colicky abdominal pain and vomiting of unknown cause, alongside indices of inflammation, CT would probably be performed as a part of standard approach.”
Response: We respect the opinion of the reviewer and we realize that these conditions are very rare. Having said that, we still think that this paper has value and adds some information to the available literature. It highlights the importance of considering the option of an underlying IBD when dealing with bezoars of the intestine and it shows different treaments that can be used in these patients. Furthermore, it gives a sinoptic view of what it has already been published so far and allows the reader to get an updated quick summary on the management of these cases.
Round 2
Reviewer 1 Report
Thank you for the corrections.
Author Response
Dear reviewer,
Many thanks for Your valuable comment
Sincerely
Emanuele Sinagra
Reviewer 3 Report
- The reference section should be extensively revised: styling of the references is inappropriate
- Most of the references are outdated, hence, they should be exchanged for more recent ones
- Did the authors obtain informed consents as Ethical review and approval were waived for the study.
Author Response
Dear reviewer
Please find as attachment the response to Your report
Many thanks,
Sincerely
Emanuele Sinagra
